# Accessing mental health services for a child living with anxiety: Parents' lived experience and recommendations

**Roberta L. Woodgate** [1] *, **Miriam Gonzalez** [1], **Pauline Tennent** [2]

1 College of Nursing, Rady Faculty of Health Sciences, University of Manitoba, Winnipeg, Manitoba, Canada,
2 Centre for Human Rights Research, University of Manitoba, Winnipeg, Manitoba, Canada

* Roberta.Woodgate@umanitoba.ca

**Data Availability Statement:** All relevant data are within the paper.

**Funding:** This work is supported by a Canadian Institutes of Health Research (CIHR) Operating Grant (Grant #: CIHR MOP -119277) https://cihr-

## Abstract

### Background

Little research attention has been given to understanding the lived experience of parents who access mental health services in the context of child anxiety disorders. This paper reports on findings specific to parents' lived experience of accessing services for their child living with anxiety and the recommendations they provided for improving access.

### Methods

We used the qualitative research approach of hermeneutic phenomenology. The sample included 54 Canadian parents of youth living with an anxiety disorder. Parents took part in one semi-structured and one open-ended interview. We used a 4 staged data analysis process informed by van Manen's approach and Levesque and colleagues' framework of access to healthcare.

### Results

The majority of parents reported being female (85%), white (74%), and single parents (39%). Parents' ability to seek and obtain services was affected by not knowing when or where to access services, having to learn to navigate the system, limited availability of services, lack of timely services and interim supports, limited financial resources, and clinicians' dismissal of parental concerns and knowledge. Provider (ability to listen), parent (willingness to participate in therapy), child (same race/ethnicity as provider), and service characteristics (cultural sensitivity) influenced whether parents perceived services as approachable, acceptable, and appropriate. Parents' recommendations focused on: (1) improving the availability, timeliness, and coordination of services, (2) providing supports for parents and the child to facilitate obtaining care (education, interim supports), (3) improving communication with and among healthcare professionals, (4) the need to recognize parents' experience-based knowledge, and (5) encouraging parents to take care of themselves and advocate for their child.

irsc.gc.ca/e/193.html. Dr. Roberta L Woodgate is supported by a Tier 1 Canadian Research Chair (CRC) in Child and Family Engagement in Health Research and Healthcare (Canadian Institutes of Health Research-Canadian Research Chair - 950-231845) https://www.chairs-chaires.gc.ca/home-accueil-eng.aspx *The funders had no role in study design, data collection and analysis, decision to publish, or preparation of the manuscript.

**Competing interests:** The authors have declared that no competing interests exist.

## Conclusions

Our findings point to possible avenues (parents' ability, service characteristics) that can be targeted to improve service access. As experts on their situation, parents' recommendations highlight priority needs of relevance to health care professionals and policymakers.

## Introduction

Anxiety disorders differ from the fear and anxiety children and youth experience as part of normative emotional development and are characterized by exaggerated symptoms about real or perceived threats that last persistently over a prolonged period of time and often lead to physical symptoms, altered behaviour, and interference with daily activities [1]. The Diagnostic and Statistical Manual of Mental Disorders, Fifth Edition (DSM-5) classifies anxiety disorders as Separation Anxiety Disorder, Selective Mutism, Social Anxiety Disorder, Panic Disorder, Agoraphobia, Specific Phobias, and Generalized Anxiety Disorder [1]. Anxiety disorders are the most common psychiatric disorders in children and adolescents with a global prevalence rate of 6.5% [2]. In Canada, the incidence of anxiety disorders is increasing with professionally diagnosed anxiety in youth 12 to 24 years old doubling from 6.0% in 2011 to 12.9% in 2018 [3]. The age of onset of anxiety disorders is childhood and if left untreated, these disorders often persist into adulthood [4, 5]. Comorbidity between anxiety disorders and other psychiatric disorders is common [6, 7] and related impairment can adversely affect various domains of health and well-being (e.g., psychosocial difficulties, more severe forms of mental illness) as well as future life outcomes (e.g., vocational, family functioning) [8–10]. While it is widely recognized that early access to effective mental health treatment is key to mitigating the disease burden over the long term, only 1 in 5 Canadian youth who needs mental health services receives them [11]. Youth experiencing mental health problems lack the means to access services alone and often turn to family and friends for help [12–14]. Thus, parents play a critical role in accessing mental health services for their child [15–17].

Healthcare access is a complex concept that has been conceptualized differently across studies, placing responsibility of access on users, health services, providers, or a combination of these [18–25]. Similarly, numerous frameworks of healthcare access have been proposed [26–29]. However, one of the most comprehensive and recent is Levesque's Conceptual Framework of Healthcare Access [26]. According to this framework, access is the opportunity to reach and obtain appropriate healthcare services in situations of perceived need for care [26]. The framework: (1) highlights the influential role of both user and healthcare service characteristics, (2) considers the process of seeking and receiving care, and (3) posits that access is influenced by five dimensions of accessibility of services (approachability, acceptability, availability and accommodation, affordability, and appropriateness), five dimensions of ability of users (ability to perceive, seek, reach, pay, and engage in healthcare), and the interaction of these dimensions [26].

A large body of evidence has documented the barriers parents face when accessing mental health services for their child [30–36]. However, these studies have not focused on childhood anxiety specifically or have been non-specific in terms of mental health diagnosis. Although documenting whether parents report different barriers when accessing services across mental health conditions is critical for informing tailored approaches to improving service access and facilitating early identification and intervention [30], less research attention has been given to studying access in the context of child anxiety disorders [37–42]. Indeed, few quantitative and

qualitative studies have focused on examining barriers parents face when accessing services for a child living with anxiety [37–39]. These studies have documented both tangible (e.g., costs; lack of knowledge on how/where to access services, difficulty identifying the child's symptoms as anxiety) [38–43] and intangible barriers (e.g., stigma, anticipating professionals would dismiss concerns, limited service provision [38–41, 43]. Further, only a handful of qualitative studies have examined parents' experience of accessing services for their child living with an anxiety disorder [12, 41, 42, 44,45].

Two of these studies were conducted in the U.K [41, 42], one in Australia [12], and two in Canada [44, 45]. Leung et al. [44] recently explored the experience of parents accessing mental services for their child (6 to 17 years) with complex mental healthcare needs residing in primarily urban centres in Alberta, Canada. Using a thematic analysis approach, the authors documented that parents' experience of access was impacted by fragmented healthcare services that lacked collaboration across disciplines, a patient-centred care approach, and information about available services. In addition, parents found navigating the healthcare system difficult and experienced related distress. Over a decade ago, Boydell et al. [45] examined parents' experience of accessing mental health care for children (3 to 17 years) diagnosed with emotional and behavioural disorders residing in rural communities in Ontario, Canada. Using a thematic analysis approach, the authors identified barriers to access (e.g., stigma, costs, lack of information about available services, waiting time, shortage of human resources, distance to services) which under different circumstances also were noted to serve as facilitators [45].

While the studies conducted in Canada documented parents' experiences of accessing services for children living with complex mental healthcare needs [44] and emotional and behavioural disorders, all of which included children with anxiety disorders [45], to our knowledge, no study has specifically focused on parents of children living with an anxiety disorder. Lacking is also a comprehensive understanding of the lived experience of parents who access mental health services for their child with anxiety. Further, the nascent literature in this area suggests the experience of parents in this context and their insights for improving access remains largely unexplored. The **purpose of this** paper is to report on findings specific to Canadian parents' lived experience of accessing services for their child (10 to 22 years) living with anxiety and the recommendations they provided for improving access. The findings emerged from a larger study that explored the lived experience of Canadian youth living with anxiety. Our work adds to the emerging knowledge base as it varies from the available studies to date given, we: (1) identified factors that influence parents' perceptions of service approachability, acceptability, and appropriateness, (2) used the qualitative research approach of hermeneutic phenomenology to document parents' lived experience of accessing mental health services for their child, and (3) analyzed and situated our findings within an access to healthcare framework to facilitate identifying avenues that can be targeted to improve access.

## Methods

### Study design and framework

Hermeneutic phenomenology served as both a guiding philosophy and methodology. The philosophy, as informed by Merleau-Ponty [46] and the method, as informed by van Manen 1990 [47] proposes that our experience in the world is full of meaning and that those meanings are always interpreted and contingent [48]. In hermeneutic phenomenology, the goal is to describe the meaning of an experience including what was experienced and how it was experienced [49]. Hermeneutic phenomenology afforded us the opportunity to deepen our understanding of the way parents' experienced accessing mental health services for their child living with anxiety.

We also used Levesque et al. [26] conceptual framework of access to healthcare to situate our findings within distinct dimensions of access. The framework offers a multidimensional view of access where the abilities of individuals seeking care interact with dimensions of accessibility of services to generate access [26]. The five dimensions of ability outlined in the framework are: (1) ability to perceive a need for healthcare (the person's ability to perceive need for care which is influenced by health literacy as well as knowledge and beliefs about health), (2) ability to seek healthcare (the person's capacity to seek care and knowledge about care options), (3) ability to reach healthcare (the person's mobility, ability to leave work to reach care, and knowledge about how to physically reach facilities/providers) and, (4) ability to pay for health services (the person's ability to pay for services without financial strain), and (5) ability to engage with the health system (the person's ability to participate in the decision-making process regarding their care) [26]. The five dimensions of accessibility of services include: (1) approachability (whether the person can identify that services exists, can be reached, and have an impact on the person in need), (2) acceptability (whether the person accepts the services and deems them socially and culturally appropriate), (3) availability and accommodation (whether services can be reached physically and in a timely manner), (4) affordabilty (the person's capacity to spend resources to use services), and (5) appropriateness (the fit between services and the person's needs and includes timeliness, adequacy, and quality of services) [26].

## Recruitment

To recruit participants and facilitate capturing the complexity and variation of parental experience, the maximum variation technique of purposive sampling was utilized [50]. To identify and select information-rich cases related to our phenomenon of interest (parents of a child living with anxiety), Parents and their child living with anxiety were recruited from both clinical and community settings. In the clinical setting, participants were recruited from the wait list of a specialized anxiety clinic in Winnipeg. Invitation letters were sent to parents and youth on the waitlist through a liaison at the clinic and those who indicated interest were provided with the contact details of the research team, should they wish to receive more information. In the community setting, participants were recruited from youth centres, teen clinics, and schools via social media. To be able to participate in the study, parents must have a child (youth) with a primary diagnosis of one or more anxiety disorders (e.g., separation anxiety, social anxiety disorder, generalized anxiety disorder, and specific phobia). Parents of youth who had a confirmed diagnosis of either obsessive-compulsive disorder (OCD) or post-traumatic stress disorder (PTSD) were not included in the project as these disorders present differently with different symptoms. Recruitment was stopped once data saturation or redundancy was achieved.

## Data collection

In keeping with the hermeneutic phenomenological approach, participants were interviewed twice to allow for participant reflection, asking follow-up questions, and in-depth exploration of topics. We collected data from parents using semi-structured interviews (first interview session) and open-ended interviews (second interview session). For the first interview session, the Anxiety Disorders Interview Schedule for Children (ADIS-IV-C/P) was administered to parents. ADIS-IV-C/P provides assessment of a wide range of disorders as per the Diagnostic and Statistical Manual of Mental Disorders [51] including anxiety, mood, and externalizing behaviour disorders [52]. Impairment ratings were generated using the Clinician Severity Rating (CSR; range = 0–8; $\geq$ 4 required to assign a clinical diagnosis [52]. We used the ADIS-IV-C/P to ensure our sample included parents of youth with a clinical diagnosis of anxiety. The reliability and validity of the ADIS-IV-C/P has been established [53, 54].

For the second interview, we used an interview guide with open-ended questions developed in collaboration with the research team and further refined (e.g., language, relevance to parents' experience) through the data collection process. The guide included questions regarding parents' perspectives about: (1) what it is like for their child to live with anxiety, (2) how living with anxiety shapes daily life for their child, and (3) what hinders and what helps their child to live satisfying, hopeful, and productive lives. Demographic information was collected about the child (e.g., age, gender, school grade, information about activities the child participates in outside of school) and the parent (e.g., gender, ethnicity, marital status, education, household income, history of diagnosed anxiety).

All interviews were conducted in person at a private location that was most convenient and comfortable to participants (i.e., mainly in their homes where the interviewer and interviewee could have some privacy). Interviews were conducted by the third author (a PhD trained research associate who has qualitative research training and experience administering the ADIS-IV-C/P) with guidance from the first author. Prior to data collection, the interviewer gave participants study information (e.g., study purpose etc), explained information relating to informed consent, and answered participants' questions. The interviewer also told participants that the senior researcher leading the project focuses on improving the mental health and well-being of young people and to do that, she engages young people and families in research and uses innovative approaches (including arts-based methods) to amplify their voices. Interviews lasted from one to three hours in length. Interview strategies used included the use of probes, silence, and elicitation of example [55]. Field notes were recorded by the interviewer after each interview detailing key points of the interview (e.g., emerging themes), the context of the interview, and assumptions or biases that could potentially impact data analysis. To preserve their authenticity, all interviews were digitally recorded and transcribed verbatim.

## Data analysis

A staged, hermeneutic phenomenological approach was applied to analyze the data. Such an approach recognizes: (1) the development of meaning and knowledge that occurs within the research encounter, (2) the interactions between the researcher, the research participant, the research aims, and the broader context in which the research takes place, and (3) acknowledges the researchers' assumptions, biases, and critical reflection at every stage [56]. Data collection occurred concurrently with data analysis. Our analysis approach involved four steps: (1) reading and re-reading interview transcripts to become fully immersed in the data with attention to significant statements, sentences, or sentence clusters that were representative of the lived experience of parents' access to mental health services for their child living with anxiety, (2) grouping units of analysis into categories and themes through analytic discussions, (3) reviewing the categories and themes using an iterative and reflexive process to ensure internal validity and that the categorizations reflected the lived experiences of participants [57], and (4) further analyzing and positioning our findings within the dimensions of ability and accessibility outlined in Levesque et al.'s framework [26]. Units of meaning were identified in the interviews and field notes, grouped into categories and themes, and organized into a table of contents in Microsoft Word. We then positioned these findings within the dimensions outlined in Levesque et al.'s [26] framework of access to healthcare. Hence, the analysis was originally data-driven but also used a conceptual framework of access to facilitate contextualizing our findings. The first (a PhD trained distinguished professor known internationally for undertaking qualitative research and using innovative approaches such as arts-based methods) and second author (a PhD trained mixed methods researcher with training and experience conducting qualitative research) collaborated on all steps and discrepancies were resolved through

discussion with the third author. We used the following strategies to enhance methodological rigour: line-by-line analysis, prolonged engagement with the data, and discussion with team members during the analysis and interpretation phases. In addition, to ensure trustworthiness, we have: (1) used methodological triangulation (collected data through semi-structured interviews, open-ended interviews, and field notes) as well as theoretical triangulation (used hermeneutic phenomenology and a theoretical framework of access to healthcare), (2) allowed for member checking (we offered transcripts to all participants, allowing them to provide feedback on the findings), (3) ensured that two researchers engaged with the data and another was available to check interpretations and resolve discrepancies, and (4) provided rich description of the findings.

### Ethical considerations

Ethical approval was obtained from the University of Manitoba Education/Nursing Research Ethics Board (ethics board approval number E2011:094) as well as ethics reviews from collaborating healthcare settings. The study was conducted in line with the principles of the Declaration of Helsinki and all participants provided written and oral informed consent (an ongoing process of verbal consent was also part of the project). Participants (both youth and parents) received an honorarium in appreciation of their time and expertise.

## Results

### Participants

A total of 54 parents living in Winnipeg, Canada participated in the study. The majority of parents reported being female (85%), white (74%), and having completed some post-secondary education (83%). More than one third (39%) reported being single parents and more than half (59%) indicated having symptoms of anxiety or a diagnosed anxiety disorder. All parents had a child (a youth) with a primary diagnosis of one or more anxiety disorders. The youth were between the ages of 10 and 22 years and the majority were female (76%). One youth participant was going through challenging life events and we decided, in collaboration with them, that it would be better for the youth and the parent not to continue in the study. Thus, we removed the youth and parent from our sample.

### Findings

One overarching theme with five main categories were identified (see Table 1). The overarching theme is parents' lived experience of accessing mental health services for their child living with anxiety and related recommendations. The five categories fall within Levesque et al.'s [26] conceptual framework of access to healthcare: (1) ability to perceive a need for mental health (MH) services and approachability, (2) ability to seek MH services and acceptability, (3) ability to reach MH services and availability, (4) ability to pay for MH services and affordability, and (5) ability to engage with the health system and appropriateness of care. Within these categories, we present relevant sub-categories and parent-identified recommendations.

**Ability to perceive a need for mental health services and approachability.** Parents' spoke about factors that contributed to their ability to perceive need for care and shared their thoughts on the approachability of services.

*Knowing when to seek help.* Various factors contributed to knowing when to seek help. For some parents, if their child's anxiety symptoms increased in severity, changed in nature, or affected their child's functioning (e.g., social, school, family), they knew it was time to seek help. When parents had personal experiences with anxiety, they also felt better equipped to

**Table 1. Study findings: Overarching theme, categories, and parent-identified recommendations.**

| Theme | Category Levesque 2013[1] Dimensions of Access | Sub-category | Parent-Identified Recommendations |
|---|---|---|---|
| Parents' lived experience of accessing mental health services for their child living with anxiety and related recommendations | Ability to perceive a need for MH services and Approachability | • Knowing when to seek help<br>• Factors influencing service approachability (e.g., misinformation, awareness of understaffing issues, difficulty getting someone to listen) | • Educate and raise awareness about anxiety and mood disorders |
| | Ability to seek MH services and Acceptability | • Not knowing where to access services<br>• Learning to navigate the system<br>• Factors influencing service acceptability (e.g., provider type, sex, ability to listen; negative perceptions about therapy, willingness to participate in the sessions; shared race/ethnicity of child and provider) | • More help and support (e.g., information) for parents• Advertise available services• Improve service coordination (e.g., key contact person, phone number) |
| | Ability to reach MH services and Availability | • Lack of available services | • Need more therapy services for children with mood disorders<br>• Increase funding for services |
| | | • Lack of timely services | • Reduce wait times |
| | | • Lack of support during waiting period | • Need for interim supports for child• Need for family support programs• More school-based programs for child (e.g., peer-support groups, yoga, education about mood disorders) |
| | Ability to pay for MH services and Affordability | • Financial resources | • Need to make services accessible to all (e.g., low income families) |
| | Ability to engage with the health system and Appropriateness | • Dismissal of parental concerns and knowledge<br>• Factors influencing service appropriateness (e.g., fit between services and family's needs, power given to youth to make decisions, lack of improvement in child's condition, cultural sensitivity of services) | • Need for experience-based knowledge to be recognized• Need for better communication between parents and healthcare professionals• Need for better communication among professionals |

Note. MH = mental health. [1]Refers to Levesque's (2013) conceptual framework of access to healthcare.

recognize their child's symptoms and seek help. However, other parents admitted to not being able to recognize when help was needed. As one parent put it: *"It got to a certain point where it's been going on and on for months and months and months before realizing there's a bigger problem" (Mother 04; Mother and Father of 16year old female youth).*

*Factors influencing service approachability.* In terms of approachability, parents were aware of and spoke about factors that they believed made it difficult to get services such as misinformation as to where to get help, understaffing issues, and difficulty getting someone to listen. For instance, some parents indicated being aware of understaffing issues in the mental health care system and how this made it challenging to get needed care. One mother shared: *"We are really understaffed in mental health care for kids, really desperately understaffed. . . which makes it harder to get the care you need. We are shorthanded in every respect" (Mother 47; Mother of 18year old female youth).*

Parents also recounted their experiences when approaching their family doctors regarding the child's condition. While some parents indicated that their first point of contact with a primary care physician went well, *"He was supportive; He put in on Prozac for his anxiety and put in a referral."(Mother 06; Mother of 10year old male youth)*, many others reported being frustrated with having to convince their doctors to listen. As one mother described it:

> *"Our experience is that starting with our own family doctor, we had to justify that we needed the help. We had to ask for the help and justify the help. We had trouble getting the family doctor to buy-in which is a prerequisite of, that your family doctor has to be involved."* *(Mother 07; Mother of 16year old female youth)*

Another father noted that: *"The biggest hurdle we've had with any kind of anxiety is getting someone to listen and take it seriously. Take it seriously that this is a problem." (Father 03; Father of 12 year old male youth).*

**Ability to seek mental health services and acceptability.** Parents' ability to seek services depended on their knowledge of care options and their ability to navigate the system.

*Not knowing where to access services.* Many parents were not aware of the health care options available for their child. As one parent expressed: *"I just found out about this thing [referring to services] that we didn't even know we had access to" (Mother 07; Mother of 16year old female youth).* Parents were aware that other families do not know where to get help either: *"Many families don't know that there's help you know. . .that there are places they can go and look into for more help." (Mother 01; Mother of 12year old female youth).* Other parents noted that their education and prior experience with the mental health system contributed to their knowledge and ability to seek help for their child: *"I know where to go. I'm lucky that way. I've had issues before with anxiety and depression." (Mother 44, Mother of 11year old youth).* However, even when parents contacted a health care provider, they encountered information gaps. One mother described this experience:

> *"You're kind of almost stopped at the door so to speak. If you go to your child's primary doctor and they don't do anything about it, and then, you're not aware of other available resources."* *(Mother 46; Mother of 11year old female youth)*

*Learning to navigate the system.* Parents' ability to seek services was influenced by their ability to navigate the system. Parents indicated they were learning to navigate the system (e.g., where services where provided such as hospitals, schools, private clinics). They often described the system as *"unnavigable" (Mother 43; Mother of 11year old male youth)* and their navigation as "hitting brick wall after brick wall" *(Mother 41; Mother of 16year old female youth).* For some parents, the persistent or fluctuating nature of their child's anxiety required the "re-navigation" of services or re-learning as needed. One parent described the experience:

> *"It's just knowing and learning what the variety of options are. If [daughter] all of a sudden next year starts to have some extra problems, do I call up the clinic that we saw? Do I have to get a re-referral? Where do I go? I don't know." (Mother 21; Mother of 1 year old female youth)*

*Factors influencing service acceptability.* Provider characteristics such as provider type (e.g., professional, students in training), provider sex (e.g., same sex as child), and provider ability to actively listen to parents and take their concerns seriously played a role in whether parents perceived services as acceptable. As one mother expressed:

*"I wanted to switch because he wouldn't take her bowel issues seriously. I thought: 'Okay that's not right, there is something wrong. Kids don't do that'. . . So then I found Dr. "S". She's awesome, but I went also because "L"[child], she didn't like that male doctor anymore either and she told me she wanted a female doctor." (Mother 08; Mother of 11year old female youth)*

Another mother shared her disappointment when her case was assigned to an intern:

*"But we did see a lady at {clinic name} for our intake meeting. She just got what we were going through. I was hoping she'd get our case and it was this guy instead. And he's an intern, but they probably look at us and go: 'Middle class, they're not suffering too bad. We'll give them to the intern' You know?" (YVA07 Mother 07, Mother of 16year old female youth)*

Parents also indicated that other factors influenced whether parents perceived services as acceptable: negative perceptions or attitudes about therapy, willingness to participate in the sessions, and whether the child shared the same race/ethnicity as the provider and thus felt at ease. As one mother explained:

*"Her father has a hard time seeing improvement. . . He should come with us to her psychology sessions. For him it's weird. He finds it very weird. . .And yeah, her psychologist is really good and she [child] likes her. [of the psychologist] She's young, she's Jewish. And she [child] is very Jewish culturally you know. So this is familiar. And she feels very comfortable." (Mother 05; Mother of 17year old female youth)*

**Ability to reach mental health services and availability of services.** Parents spoke about the availability of mental health services and their experiences reaching services in a timely manner.

*Lack of available services.* Lack of services was common throughout parents' narratives. One mother stated: *"The [clinic] closed and lost her file. I had phoned a bunch of anxiety groups but they were all kind of adult self-led groups, I couldn't find anything for youth."(Mother 54: Mother of 16year old female youth)*. Parents also shared that they were advised to contact crisis services if needed as there were no available outpatient services. For some parents, this was the only way to obtain care. One father noted: *"Without those [crisis] events, it is difficult to get help." (Father 53; Father of 13year old female youth)*.

*Lack of timely services.* A lack of timely services due to referrals and waitlists was also a nearly universal experience. In the Canadian context where this study took place, access to mental health specialists usually requires a referral from a primary care physician. Parents expressed that securing a referral was not easy or straightforward. One mother shared:

*"But yeah, I mean doctors should be able to do more. I have asked her doctor to send her to a psychiatrist and her doctor refuses. She said she doesn't need it because it's her job as the doctor to prescribe the medications." (Mother 54; Mother of 16year old female youth)*

Further, delays due to waitlists were very disappointing as many families had already experienced a lengthy wait due to referrals. One mother shared her experience:

*"He is not going to school and I'm phoning everybody in the phonebook, everybody that I can think of to try and get some immediate help. I need somebody to help me right now. And everybody has a waiting list and it takes forever and we had to wait. First off, we had to get a*

*whole bunch of referrals to get him assessed. And then, once he was assessed, there was six months before the anxiety clinic even phoned us.*"

(Mother 43; Mother or stepfather of 11year old male youth)

Parents expressed frustration with the long wait times and the process of getting their child assessed. One parent shared:

"*I guess for us what's been frustrating is, you go, and you tell the doctor what's happening with your child and they don't really say, well your child has this or your child has that. It's a very slow process just to get her assessed.*" *(Mother 04; Mother of 16year old female youth)*

The delay in care was also distressing for parents who witnessed their children continue to suffer. Many parents described feeling helpless in this regard:

*The waiting lists are very difficult, especially when you see them suffering socially and at school and there's nothing you can do. And it's like, the more time that goes by, the more ingrained this is going to be with her. And the older she is, its going to be harder to deal with.*"

*(Mother 08 Mother of 11year old female youth)*

Parents also indicated that having to wait for a few months to see a specialist often meant accessing services when the child's problems had subsided. As one mother shared:

"*And so from April, when we finally got the process going, we were seen in August. Well, in August she's not in school. We went to finally see the psychiatrist and he's saying, 'So, I'm hearing you're having some difficulties.' She goes, 'Nope. No bad dreams, no issues with friends, nothing,' because it's summer.*" *(Mother 21; Mother of 13year old female youth)*

*Lack of support during waiting period.* In addition, parents indicated there were no interim supports while they waited for services. Although some parents tried to manage on their own (e.g., doing research, getting books), they reported feeling *"woefully ill-equipped" (Mother 30; Mother of 17year old female youth)* to support their child while they waited for care. As one mother described it:

"*I just got a book from the library. That was actually the last time I phoned the [specialized clinic]. She suggested some books. So I got them, books for teenagers, but I don't know how to approach it. And with her, I still am undecided about if it's helpful or not. I'd rather have the professionals' help in this.*" *(Mother 07; Mother of 16year old female youth)*

**Ability to pay for mental health services and affordability.** Parents also spoke about the cost of services and how financial resources served as enablers and barriers to accessing services in the private sector and expressed concern for families with less financial resources.

*Financial resources.* Having financial resources meant being able to afford accessing timely care via the private sector. Some parents with financial resources often chose to pay for private mental health care in lieu of, or while waiting for, publicly funded services. One mother described the reasons that prompted her to take this route:

*Well, because if you don't deal with it, if I had waited until September/October- I know there will be no service over the summer, my daughter probably wouldn't be in school or in*

*gymnastics or anything if I had not got help. I'm spending three hundred dollars a month on getting her help, but I mean it's worth every penny."* (Mother 02; Mother of 11year old female youth)

However, parents were aware that private services were not affordable for every family (those with less financial resources) and expressed concern for children of these families. As one mother shared: *"I have the means to pay for a psychologist but other people may not and that would be devastating."* (Mother 02; Mother of 11year old female youth). Another parent stated: *"I would hate to think of what happens to a kid like ours, who didn't have two people working full-time to save her."* (Father 07; Father of 16year old female youth). Indeed, paying for services in the private sector was challenging for parents lacking financial resources (e.g., private insurance, disposable income). One father described it as follows:

*"Just to reach the medication stuff you need a psychiatrist who can put you on medication. Just to get into a psychiatrist you either pay yourself or you go through a [public] system. Pay yourself, we don't have the chance. So we have to go through the system and to go through the system, it takes a year after a major breakdown."* (Father 53; Father of 13year old female youth)

Even for families with private health insurance, the coverage was usually insufficient. These families indicated they could not afford the entire cost of private care:

*"When we saw that doctor that just basically spent the whole time telling her, 'You don't have to be here,' we paid $260 for that and that was half of our coverage for the year. So, (chuckle) we can't afford it, we can't afford to get much private help."* (Father 07; Father of 1 year old female youth)

**Ability to engage with the health system and appropriateness.** Parents also spoke about their experiences engaging with healthcare professionals and factors that influenced whether they perceived services as appropriate.

*Dismissal of parental concerns and knowledge.* Upon approaching health care providers, some of the parents in our study were faced with dismissal. Many reported experiences of healthcare professionals not taking their first-hand experiences of their child's symptoms seriously:

*"They don't listen to parents. . .And you can kind of see they're thinking: 'Oh, you're overprotective' or 'Oh, you've caused this problem.' I think there's a lot of stuff going through people's heads that's making them dismiss what the parents have to say. . .And they don't get that we live with these kids."* (Father 07; Father of 16year old female youth)

In addition, parents shared that their knowledge of their child and parental expertise was not recognized or valued by healthcare professionals. As one mother stated:

*"I'm big on those people understanding that we know our kids better than anything. And so even though you're a professional and you've read all the books and you've seen it a million thousand times, my spin on it, or what I can tell you is going to be unique and respecting that information."* (Mother 09; Mother of 12year old female youth)

*Factors influencing service appropriateness.* The fit between services and the family's needs, the power afforded to youth in making decisions about their health, lack of improvement in the child's condition, and the cultural sensitivity of services influenced whether parents perceived services as appropriate. Parents shared that often, services were available but were not the "right" ones. For instance, one mother shared her experience as she was referred to group therapy despite feeling this was not an option:

*"Well, they wanted to refer us out of [specialized service]. . .. So they wanted to refer us to another generic program and we didn't feel we were having a lot of success with that type of program. So we thought,' Well, let's wait and try to get to a place that's going to be helpful'."*

*(Mother 07; Mother of 16year old female youth)*

Having to meet with healthcare professionals in a format that was not ideal to parents was also common. As one father shared:

*"And I phoned that nurse and I said, 'We would like to meet with you separately. We do not want to sit in front of her and do all this, we don't work that way'. We have to all sit in the same room. So I said: 'This is damaging to her, we, we don't want to do this'." (Father 07; Father of 16year old female youth)*

Some parents questioned the appropriateness of services based on lack of improvement or progress:

*"And also with these programs, I don't think there's an accountability. There's a lot of people {psychiatrists} patting themselves on the back with how much good they're doing for us and telling us: 'Oh yeah we're making really good progress and everything' and we would see no progress, we didn't see any difference." (Mother 07; Mother of 16year old female youth)*

Other parents questioned the appropriateness of letting youth make all the decisions. As one mother shared: *"She doesn't see the need for help or understand the need for help, but she's the one being given that decision. We're giving them a lot of power to make decisions about something they don't understand." (Mother 07, 16year old female youth).* Yet others questioned the cultural appropriateness of services for various groups:

*"I used my resources because we can pay. They [services/the system] don't block us but they would block many people. And then I don't even know how you would layer onto that culturally appropriate, or if you were from another ethnicity or background or spoke a different language, I mean, I don't know." (Mother 34; Mother of 17year old female)*

## Parent-identified recommendations

As parents shared their experiences of accessing services for their child living with anxiety, they also spoke about ways to facilitate accessing services. Table 2 lists the recommendations identified by parents with representative quotations.

**Ability to perceive a need and seek services.** To enhance parents' ability to perceive a need for mental health services and their ability to seek those services, parents noted that education and more awareness about anxiety and mood disorders was needed. They also expressed wanting more help and support (e.g., information) and highlighted the need to advertise available

Table 2. Parent-identified recommendations for improving service access by dimension of access.

| Category (Levesque 2013[1] Dimensions of Access) | Parent-Identified Recommendation | Representative Quotation |
|---|---|---|
| Ability to perceive a need for MH services (Knowing when to seek help) | • Educate and raise awareness about anxiety and mood disorders | *"Education, I think if people could just really get more awareness about anxiety, that would help future parents. . . I think it's also important to talk to other people that are experiencing it."* (Mother 09; Mother of 12year old female youth) |
| Ability to seek MH services (Not knowing where to access services; Learning to navigate the system) | • More help and support (information) for parents | *"More support and information for parents who know nothing about it, you know, you try to do so much on your own, but maybe you need more help from the professionals."* (Mother 01; Mother of 12year old female youth) |
| | • Advertise available services | *"But they [services} have to be well-advertised, so when a person, when you're at your wits end, you go, 'Oh I saw this poster, I can phone this number'."* (Father 07; Father of 1 year old female youth) |
| | • Improve service coordination (e.g., a phone number or key contact person) | *"I'm wondering if there wouldn't be a role for like a case manager but an external case manager. I guess in a perfect world they would talk to you and say to you how's it going. . .And then you could say, well we don't know, or nothings progressing or is the same, and then they could perhaps I don't know how they would do it but somehow you know refocus the people that are trying to help or you know send you to a different service or something."* (Father 07; Father of 16year old female youth) |
| Ability to reach MH services (Lack of available services; Lack of timely services; Lack of support during waiting period) | • Need more therapy services for children with anxiety and other mood disorders | *"It would be nice to see more programs and services for kids with mood disorders"* (Mother 08; Mother of 11year old female youth) |
| | • Increase funding for services | *"And "And I understand that all these wait lists and all these procedures are in place because there's not enough spots, there's not enough services, but that's where we need to be putting all of our money."* (Mother 47, Mother of 18year old female youth) |
| | • Reduce wait times | *"If it was somehow organized better to make sure you're getting the help more quickly. I think it would cost a lot less money to the system."* (Father 07; Father of 16year old female youth). *"Because I know the waitlists for the anxiety clinic, there's a long waitlist which I'd like to see that improved in all our services."* (Mother 02; Mother of 11year old female youth) |
| | • Need for interim supports | *"And if the wait has to be six months, have a plan for the interim to help that parent along."* (Mother 02; Mother to 11year old female youth) |
| | • Need for family support programs | *"Maybe they could provide support to the family. There should be support to the family, support programs . . ."M" said no [to speaking to a professional] and I said: 'Well, what about us? Can we come and see the social worker?'. . .'No, only if she comes'".* (Father 07; Father of 16year old female youth) |
| | • More school-based programs for child (e.g., peer-support groups, yoga, education about mood disorders) | *"I think that there should be more awareness programs in the schools, in the curriculum, just to talk to the kids about things, like about anxiety is or what bipolar is or ADHD. . . .Or whatever the issues are, just so the next generation just grows up more aware and tolerant and accepting of people."* (Mother 08; Mother of 11year old female youth) |
| Ability to pay for MH services (Financial resources) | • Need to make services accessible to all (e.g., low-income families) | *"Yeah I think if there was maybe more things that were more affordable to people that don't have a lot of money. . .And where "L" could be around kids that are more like her that understand. I know there are some programs but some of them, like I said, are either in horrible neighbourhoods or they're pricey."* (Mother 08; Mother of 11year old female youth) |

*(Continued)*

**Table 2.** (Continued)

| Category (Levesque 2013[1] Dimensions of Access) | Parent-Identified Recommendation | Representative Quotation |
|---|---|---|
| Ability to engage with the health system (Dismissal of parental concerns and knowledge) | • Need for experience-based knowledge to be recognized<br>• Need for better communication between parents and healthcare professionals<br>• Need for better communication among professionals | *"They [healthcare providers] need to listen to us but they also need to give us back some information." (Father 07; Father of 16year old female youth)*<br>*"There needs to be a bit more communication. I said something like, 'You know we haven't seen a lot of change other than the medication' and the nurse said right away, 'Well, "M's" done a lot of work too you know', and that's probably true but we don't know what work she's done." (Mother 07; Mother of 16year old female youth)* |

Note. MH = mental health. [1]Refers to Levesque's (2013) conceptual framework of access to healthcare.

services (see Table 2). Other parents suggested coordinating services more effectively. More specifically, parents suggested that having a central place, a phone number or a key contact person, they could call for information about resources and services would be helpful. One mother suggested:

> *"There needs to be one phone number. If you're having trouble with your teen, phone this number and then there should be a worker or somebody that they see you through and help you find the right agency. So they were telling me it's all integrated, well it's not, they have the same file but that's about it. . . I think people in the system think it's a certain way but when you're outside the system it's totally different than what they think it is. . . . So, yeah, if there was one agency, you phone this number, they help you." (Mother 07; Mother of 16year old female youth).*

**Ability to reach services.** For parents to reach services, mental health services need to be available. Parents highlighted the need for more services and for allocating money to mental health services to this end. One mother expressed:

> *"And I understand that there's not enough spots, there's not enough services, but that's where we need to be putting all of our money. Because these are kids whose life patterns aren't formed yet right. These are children who are young enough still that if you give them the help they need, they will become healthy adults and will not need services for the rest of their lives."*
>
> *(Mother 47, Mother of 18year old female youth)*

Parents also suggested decreasing the time they wait for appointments, through better organization or coordination, to ensure reaching services in a timely manner. As one parent noted: *"If it was somehow organized better to make sure you're getting the help more quickly. I think it would cost a lot less money to the system." (Father 07; Father of 16year old female youth).* Many parents also called attention to the urgent need for interim supports, for their child and themselves, as they waited for care. For example, some parents suggested the need for more school-based programs for the child such as peer-support groups, education about mood disorders, and extra-curricular programming such as yoga. Parents also highlighted the need for family support (see Table 2 for representative quotations).

**Ability to pay for services and engage with the health system.** Parents also recommended having mental health services that are accessible (affordable) to families with less financial resources. They also stressed the need for healthcare professionals to recognize their

**Table 3. General recommendations provided by parents for parents.**

| Topic | Parent-Identified Recommendation | Representative Quotation |
|---|---|---|
| Getting help | • Get help and keep trying to get the help you need | *"You need to use the system and get in there and use it. Get in there!. . . And keep trying to get help because you have to keep trying."* (Mother 07; Mother of 16year old female youth) |
| | • Educate yourself about anxiety (e.g., get books, do your research) | *"Take it seriously. Get books. Understand anxiety."* (Mother 02; Mother of 11year old female youth) |
| | • Figure out if you can help your child. If not, reach out and get help | *"Try to figure out if you're equipped to deal with the situation or not and be very honest with yourself about it. If you're not equipped, find help somewhere else. . .And don't try to solve a problem if you're not equipped to do it."* (Mother 05; Mother of 17year old female youth) |
| | • Don't give up | *"Just don't ever give up."* (Father 03; Father of 12year old male youth) |
| Self-care | • Try not to do it all yourself | *"Don't try to do it all yourself. Try to get help."* (Mother 01; Mother of 12year old female youth) |
| | • Don't be hard on yourself | *"And be patient and don't be hard on yourself."* (Mother 06; Mother of 10year old male youth) |
| | • Don't feel guilty | *"Don't worry about it so much, don't feel so guilty, cause you're doing the best you can. . .And remind yourself of that."* (Father 06; Father of 10year old male youth) |
| Helping the child | • Be your child's advocate | *"You have to advocate for your kids."* (Mother 04; Mother of 16year old female youth) |
| | • Be encouraging and comment on your child's accomplishments | *"So you know being that sounding board [for your child], reflecting, encouraging, but doing it in a realistic manner."* (Mother 02; Mother of 11year old female youth) |
| | • Let your child know he/she is loved | *"Let them [kids] know they are loved."* (Father 03; Father of 12year old male youth) |
| | • Be understanding, patient, and supportive | *"Use your time and patience as a tool, just be prepared that you're going to take that one step ahead and then you might take five steps back, but understanding that those little steps for, for anybody dealing with that, are like super you know climbing over a mountain steps."* (Mother 09; Mother of 12year old female youth) |

experience-based knowledge and the need for better communication between parents and healthcare professionals as well as among professionals. As one mother expressed:

> *"I think communication needs to be better. . .Obviously you know, the counsellor, the principals, the social worker all know "A" and her struggles. But I found the last couple of years that I've been the one that has let the next new teacher know about what's going on and know how serious it is."*
>
> (Mother 09; Mother of 12year old female youth)

In addition, parents shared advice for other parents. These ranged from encouraging parents to educate themselves about anxiety, taking care of themselves, and advocating for their child (see Table 3).

## Discussion

The objective of this paper was to report on findings specific to Canadian parents' lived experience of accessing services for their child (10 to 22 years) living with anxiety and the recommendations they provided for improving access to services. Given that research on parents' experience of accessing mental health services specific to child anxiety disorders is limited, this study adds to the emerging evidence in this regard. That is, we have highlighted areas situated within a framework of access to healthcare [26] that can be targeted for improving access in this context: parents' ability to seek and obtain care (e.g., ability to perceive a need for services, ability to seek services) and service characteristics (e.g., availability, affordability). We have

also identified provider (e.g., ability to listen), parent (e.g., willingness to participate in therapy), child (e.g., same race/ethnicity as provider), and service characteristics (e.g., cultural sensitivity) that influenced whether parents perceived services as approachable, acceptable, and appropriate. Parents' thoughts on service approachability, acceptability, and appropriateness were not examined nor reported in the handful of studies of parents' lived experience of accessing services for childhood anxiety. Finally, we report on recommendations parents provided for improving access which focused on: strengthening parents' own ability to seek and obtain care (e.g., education about mood disorders, information about services), improving the availability of services (e.g., more services and enhanced coordination), facilitating parents' engagement with the health system (e.g, recognition of their experience-based knowledge), and providing advice for parents in similar situations (e.g., take care of themselves, advocate for the child).

## Ability of parents to seek and obtain care

For many parents in our study, their ability to seek and obtain care was hindered by not knowing when to seek help and where to access services, having to learn to navigate the mental heath care system, facing healthcare professionals' dismissal of their concerns and knowledge, and having limited financial resources to obtain services in the private sector or those not covered by insurance plans. Our findings are consistent with: (1) results of the handful of studies that have documented parents' lived experience of access to mental health services for child anxiety disorders both in Canada [44, 45] and abroad [12, 41, 42]; (2) the few studies that have examined barriers to access in the context of child anxiety disorders [38–42]; and (3) the literature on the barriers parents face when accessing mental health services (not specific to anxiety) for their child [30–36]. The finding that the barriers related to parents' ability to obtain care identified in this study have been reported in other studies as noted above highlights the need to support parents, regardless of mental health condition for which care is sought, by strengthening their ability to identify mental health problems and ensuring information about available services and mental health care system navigation is readily available (e.g., primary care clinics). Parents also reported that healthcare professionals' dismissal of their concerns and experience-based knowledge were factors that negatively impacted their ability to obtain care. This highlights the importance of clinician sensitivity, actively listening to parents' concerns, and engaging parents throughout the process of obtaining and receiving care. Finally, the financial barriers to obtaining services noted by parents point to the need for solutions that will benefit all segments of the population such as increasing publicly funded access to psychotherapies [35, 58] or strengthening service delivery at the primary care level [35]. Although initiatives to expand access to publicly funded psychotherapy and strengthen primary care's ability to facilitate access through stepped care models are underway in the Canadian provinces of Ontario, Quebec, and BC [35], concerted efforts in this area are needed in Manitoba [59].

## Characteristics of mental health services

Parents in our study spoke about their access experience in relation to service approachability, availability, acceptability, and appropriateness of services (parents spoke about the cost or affordability of services in the context of financial resources as enablers or barriers to access as outlined in the previous section). Service and provider characteristics influenced parents' perceptions of service approachability. That is, parents spoke at length about how lack of information of available services, staff shortages, and difficulty getting someone to listen made it challenging to recognize that services for their child could be reached. Although studies of parents' lived experience of accessing services for childhood anxiety documented these

challenges [12, 41, 42, 44, 45], our study is the first to report the influential role of these challenges on the approachability of services (whether information about services is available/accessible and parents know that services exist and can be reached) [26].

The poor availability of services was also a common theme in our study as was lack of timely services and interim supports. These findings are consistent with results of qualitative studies that have examined parents' lived experience of access in the context of child anxiety disorders [12, 41, 42, 44, 45], studies that have focused specifically on access to services for childhood anxiety [38–40], and the larger body of evidence on barriers parents face when accessing mental health services for their child (non-specific to anxiety or a mental health diagnosis) [30–36]. Our findings thus reinforce that for childhood anxiety as well as other childhood mental health disorders, service availability (whether services are physically available and can be reached in a timely manner) [26] necessitates ongoing investment and enhancement.

Parents in this study also spoke about factors that influenced their perception of the acceptability of services: (1) provider type (professional as opposed to trainee), sex (same sex as child), and ability to listen, (2) parents' perceptions (positive as opposed to negative) about therapy and willingness to participate in therapy sessions, and (3) shared race/ethnicity of child and provider. They also spoke about factors that influenced whether they perceived services as appropriate: child-related factors such as lack of improvement in the child's condition and power given to the child to make decisions, and service-related characteristics such as the cultural sensitivity of services and the fit between services and family's needs. Of the five studies that have documented parents' lived experience of accessing services for childhood anxiety [12, 41, 42, 44, 45], none identified factors that influenced parents' perception of acceptability and appropriateness of services. Only in Boulter and Rickwood's study [12], the fit between services and family's needs was identified as having an impact on future help-seeking but not on perceptions of service appropriateness or acceptability. Further, other studies of access to services for childhood anxiety have focused primarily on identifying barriers and facilitators to access [38–40]. Thus, in the context of accessing services for children living with anxiety, our findings are important and point to factors that should be considered to ensure provision of appropriate and acceptable care.

## Parent-identified recommendations

Parents' recommendations focused on: (1) improving the availability, timeliness, and coordination of services, (2) providing supports for parents to facilitate seeking and obtaining care, (3) providing interim supports for the child (e.g., school-based programs such as yoga) and parents (e.g., support groups), (4) improving communication with and among healthcare professionals, and (5) the need to recognize parents' experience-based knowledge. To our knowledge, few studies have reported parent-identified recommendations for improving access to services for childhood anxiety. Of the handful of studies that have documented parents' lived experience of access to mental health services in this context [12, 41, 42, 44, 45], only two studies detailed parents' recommendations for improving service access [44, 45]. Of the studies that have examined parents' access to services for child anxiety disorders [38–42] and of studies that have documented parents' access to mental health services for their child (non-specific to anxiety) [30–36], only Cohen et al. [36] reported parents' recommendations for supporting parents in obtaining care for their children. The recommendations provided by parents in our study echo the recommendations provided by parents in those few studies that documented parents' recommendations [36, 44, 45]. Our findings reinforce that irrespective of childhood mental health disorder, parents want accessibility to mental health services (e.g., availability of services, reduced wait times), improved coordinated care (e.g., one person or facilitator as a

contact through the care journey, better communication among health care professionals), engagement of child and parents in the decision-making process (e.g., better communication between parents and clinicians), as well as education (e.g., about anxiety and mood disorders) and support (e.g., information about available services).

In addition, parents in our study advocated for (1) the need for funding to increase the availability of services, (2) making services accessible to all segments of the population (e.g., families with limited financial resources), and (3) for parents to take care of themselves and advocate for their child (provided recommendations for other parents). For parents in our study, accessing services was challenging. Thus, advocating for funding to increase available services is not surprising. Parents in our study were aware that for families with less financial resources, accessing timely care via the private sector may be difficult and noted that services should be accessible to everyone. Further examination of parents' recommendations for making service access equitable and inclusive is warranted. Finally, an unexpected finding was that parents provided recommendations for other parents that spanned from encouraging them to take care of themselves to tips for supporting their child. Having the opportunity to provide advice for others in similar situations may have been therapeutic for parents in our study and highlights the importance of support groups (e.g., family, friends, peers) for ensuring parents feel heard and less alone.

## Implications

Firstly, we identified parent, child, provider, and service characteristics that influenced parents' perceptions of service approachability, acceptability, and appropriateness in the context of service access for childhood anxiety. The engagement of parents and their children in research about service access and in identifying solutions to the access-related challenges they encounter is critical to ensuring services are relevant to their needs (appropriate), meet their expectations (acceptable), and are available when and were needed (accessibility).The influence of other provider, parent, child, and service characteristics on parents' perceptions of service approachability, accessibility, and appropriateness needs to be further explored. Second, our findings reinforce the need to support parents by widely disseminating accessible, public health information about: (1) anxiety and other mood disorders (e.g., symptoms, how to identify), (2) when to seek help, (3) available services and supports (in-person or online self-help or peer supports), and (4) navigating the mental health care system. Promoting public awareness in this regard is important especially now as we move through waves of the COVID-19 pandemic. Third, the poor availability of services parents reported highlights the need for continued service enhancement efforts [35, 58, 59]. Finally, our findings also highlight the importance of ensuring clinicians (e.g., primary care providers, pediatricians): (1) have information about and can point parents to resources in the community and/or hospital as per above, (2) engage parents and the child as much as possible in the process, and (3) actively listen and recognize parents' experience-based knowledge.

## Limitations and future directions

Parents in this study were mostly white, female, living in an urban setting, had some level of post-secondary education, and had a previous history of mental health problems. Future research should more closely examine the perspective of caregivers who are male, live in rural settings, and are from diverse professional and cultural backgrounds. For the majority of parents in our study, accessing services was challenging. Future research on the journey of parents who have a positive experience in accessing services for their child would be informative. The recommendations parents provided focused on improving their ability to access care

(perceive a need, seek and reach services, pay for services, and engage in health care) and availability and affordability of services. Our interview guide did not include questions specific to the conceptual framework but asked parents to speak about what would help their child living with anxiety. Future research exploring parents' recommendations for improving other accessibility dimensions such as the acceptability, appropriateness, and approachability of services is warranted. Finally, more than half of the parents in this study reported symptoms or a diagnosis of anxiety. Future research on the impact of having a child living with anxiety on parent's own mental health is warranted.

## Conclusions

By studying parents' lived experience of accessing services for their child living with anxiety using hermeneutic phenomenology, we were able to: (1) point to factors that should be considered to ensure services are approachable, acceptable, and appropriate from parents' perspectives, (2) highlight the importance of supporting parents by strengthening their ability to seek and obtain care, and (3) reinforce the need to ensure services are available at the right place and time.

## Supporting information

**S1 File. Consolidated criteria for reporting qualitative studies (COREQ) checklist.** (DOCX)

## Acknowledgments

The authors are grateful to all the participants who shared their experiences with us.

## Author Contributions

**Conceptualization:** Roberta L. Woodgate.

**Data curation:** Roberta L. Woodgate, Pauline Tennent.

**Formal analysis:** Roberta L. Woodgate, Miriam Gonzalez.

**Funding acquisition:** Roberta L. Woodgate.

**Investigation:** Roberta L. Woodgate, Pauline Tennent.

**Methodology:** Roberta L. Woodgate.

**Project administration:** Roberta L. Woodgate.

**Resources:** Roberta L. Woodgate.

**Software:** Roberta L. Woodgate.

**Supervision:** Roberta L. Woodgate.

**Validation:** Roberta L. Woodgate.

**Visualization:** Roberta L. Woodgate, Miriam Gonzalez.

**Writing – original draft:** Roberta L. Woodgate, Miriam Gonzalez.

**Writing – review & editing:** Roberta L. Woodgate, Miriam Gonzalez, Pauline Tennent.

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
