## [Decision Letter · Decision Letter 0]

27 Feb 2023

PONE-D-23-01298Accessing mental health services for a child living with anxiety: Parents’ lived experience and recommendationsPLOS ONE

Dear Dr. Woodgate,

Thank you for submitting your manuscript to PLOS ONE. After careful consideration, we feel that it has merit but does not fully meet PLOS ONE’s publication criteria as it currently stands. Therefore, we invite you to submit a revised version of the manuscript that addresses the points raised during the review process.

We look forward to receiving your revised manuscript.

Kind regards,

Nelsensius Klau Fauk, S.Fil., M., MHID, MSc, PhD

Academic Editor

PLOS ONE

Journal Requirements:

Additional Editor Comments:

Please carefully address the reviewers' comments and resubmit.

Reviewers' comments:

Reviewer's Responses to Questions

**Comments to the Author**

1. Is the manuscript technically sound, and do the data support the conclusions?

Reviewer #1: Yes

Reviewer #2: Yes

2. Has the statistical analysis been performed appropriately and rigorously? 

Reviewer #1: N/A

Reviewer #2: N/A

3. Have the authors made all data underlying the findings in their manuscript fully available?

Reviewer #1: Yes

Reviewer #2: Yes

4. Is the manuscript presented in an intelligible fashion and written in standard English?

Reviewer #1: Yes

Reviewer #2: Yes

5. Review Comments to the Author

Reviewer #1: Comments & suggestions to the authors

This study investigated the mental health services for a child living with anxiety, experienced by the parents. The Introduction clearly mentioned the seriousness of the problem and the rest of the paper have given detailed explanation of the study. However, some development could be used:

- The terminology for anxiety disorder should refer to ICD or DSM to make a clear distinction with anxiety as a symptom

- Do update the prevalence rate of anxiety disorder among adolescence as the one cited came from 2005 (WHO has more updated data)

- It is great to read the effort of the authors to explain/describe any methods used in this paper for the learning process of the readers

- What was the reporting guideline use by the authors?

- For the recruitment, the authors used the maximum variation technique of purposive sampling. Please elaborate more.

- Data collection: better if the data collection can be divided into two parts, to make a clearer understanding of the data collection process, between the semi-structured interviews and open-ended interviews. Was the AIDS-IV-C/P used for the parents or the children? What was the diagnostic criteria of it? Who did the interview for the diagnosis? (credentials) What were the purposes of each either semi-structured interviews and open-ended interviews? Also for the open-ended interviews, who did the interviews?

- How about the trustworthiness of the study?

- Data analysis: the first and second author did the data analysis. Describe the background of the authors, including gender and experience in qualitative study.

- Results: start with the process of the data analysis (how many codes into themes?) It is good as the authors have provided the coding tree into the tables.

- Overall, it is an interesting study for a better health care services and policy.

Reviewer #2: The paper reports very important findings. It is well written and easy to follow. I have only a few minor points on the method section to be addressed.

Please used the COREQ checklist to guide the report of the method section.

What is the qualification and research experience of the research assistant who conducted the interviews?

Who took the field notes during the interviews? Was it the RA/interviewer of somebody else? If the same person did the interviews and at the same time took notes, then any reflections of the difficult experience? It is not easy to interview and probe information from a participant if the interviewer has to take notes as well.

Who did the transcription of the audio recordings? Was it the RA or one of the authors? If it was not the RA, then how did you integrate the field notes into the transcripts?

6. PLOS authors have the option to publish the peer review history of their article (what does this mean?). If published, this will include your full peer review and any attached files.

Reviewer #1: No

Reviewer #2: No

---

## [Author Response · Author response to Decision Letter 0]

7 Mar 2023

We thank reviewers for their insightful comments. We believe our manuscript has benefited from your suggestions.

Reviewer #1: Comments & suggestions to the authors

This study investigated the mental health services for a child living with anxiety, experienced by the parents. The Introduction clearly mentioned the seriousness of the problem and the rest of the paper have given detailed explanation of the study. However, some development could be used:

- The terminology for anxiety disorder should refer to ICD or DSM to make a clear distinction with anxiety as a symptom

We have added a definition of anxiety disorders using the DSM-5. Please see track changes on p. 3, lines 66-72. We have added the new reference citation (see Reference List) and adjusted all reference citations accordingly (in-text and reference list).

- Do update the prevalence rate of anxiety disorder among adolescence as the one cited came from 2005 (WHO has more updated data)

We have provided information from a 2015 meta-analysis of the worldwide prevalence of mental disorders (including anxiety) for children and adolescents. Prior to this meta-analysis, there was no information as to a worldwide-pooled prevalence for this population specifically.

We checked WHO resources but found it challenging finding the reference citations provided (e.g, links no longer worked etc) or reference citations for anxiety disorders were not provided. For instance, for the following WHO resource: Mental health of adolescents (who.int) , although a reference citation is provided for the global prevalence of mental health conditions, the year of this source was not provided. Further, for emotional disorders including anxiety, no reference citation was provided. Thus, given the robust methodology used in the meta-analysis cited above, we decided to use that reference instead. 

- It is great to read the effort of the authors to explain/describe any methods used in this paper for the learning process of the readers. What was the reporting guideline used by the authors?

We have used the COREQ checklist. Under the “Supporting Information” section of the manuscript, we note we are including an additional file labeled “S1 File. Consolidated criteria for reporting qualitative studies (COREQ) checklist”. Please see p. 36, line 914. Also see the additional file. The document highlights where in the manuscript (page number) checklist items are reported. 

- For the recruitment, the authors used the maximum variation technique of purposive sampling. Please elaborate more.

We have noted that to facilitate capturing the complexity and variation of parental experience, we used purposive sampling and refer readers to an excellent reference citation that guided our decision regarding sampling strategy and sample size. Please see p. 7, lines 165-166

We have also added that to identify and select information-rich cases related to our phenomenon of interest (parents of a child living with anxiety), we recruited from both clinical and community settings. We then provide additional information as to recruitment approach in each setting (e.g., Clinical setting: through wait list of a specialized anxiety clinic; Community setting: youth centres, teen clinics, and schools). Please see p.7, lines 168-173.

- Data collection: better if the data collection can be divided into two parts, to make a clearer understanding of the data collection process, between the semi-structured interviews and open-ended interviews. Was the AIDS-IV-C/P used for the parents or the children? What was the diagnostic criteria of it? Who did the interview for the diagnosis? (credentials) What were the purposes of each either semi-structured interviews and open-ended interviews? Also for the open-ended interviews, who did the interviews?

We have divided that paragraph into two paragraphs. Please see p. 7, lines 180-198. We hope this facilitates readers’ understanding of the data collection process.

In the first paragraph we provide information about the semi-structured interviews. We note that: (1) the AIDS-IV-C/P was used with parents, (2) we used the Clinician Severity Rating (range -0-8; > 4 required to assign a diagnosis), and (3) that we used the ADIS-IV-C/P (purpose) to ensure our sample included parents of youth with a clinical diagnosis of anxiety. See p. 7, lines 183-189. 

We also specify that our findings emerged from a larger study that explored the lived experience of Canadian youth living with anxiety for which both youth and their parents were interviewed. Please see p. 5, lines 131-132. In the Methods section, we highlight the questions that guided the open-ended interviews. Please see p. 8, lines 193-196.

We also note that the third author, a PhD trained research associate who has qualitative research training and experience administering the ADIS-IV-C/P conducted all interviews with guidance from the first author. Please see p. 8, lines 201-203.

- How about the trustworthiness of the study?

We have additional information about strategies used to enhance methodological rigour and trustworthiness. Please see p. 9 and 10, lines 234-242.

- Data analysis: the first and second author did the data analysis. Describe the background of the authors, including gender and experience in qualitative study.

We have added this information. Please see p.9, lines 231-233.

- Results start with the process of the data analysis (how many codes into themes?) It is good as the authors have provided the coding tree into the tables.

Yes, we have provided the coding tree in the tables. Please see Table 1 on p. 11, Table 2 on p.20, and Table 3 on p. 24.

- Overall, it is an interesting study for a better health care services and policy.

Reviewer #2: 

-The paper reports very important findings. It is well written and easy to follow. I have only a few minor points on the method section to be addressed. Please used the COREQ checklist to guide the report of the method section.

We have used the COREQ checklist. Under the “Supporting Information” section of the manuscript, we note we are including an additional file labeled “S1 File. Consolidated criteria for reporting qualitative studies (COREQ) checklist”. Please see p. 36, line 914.

-What is the qualification and research experience of the research assistant who conducted the interviews?

We present this information on p. 8, lines 201-203.

-Who took the field notes during the interviews? Was it the RA/interviewer of somebody else? If the same person did the interviews and at the same time took notes, then any reflections of the difficult experience? It is not easy to interview and probe information from a participant if the interviewer has to take notes as well.

We note that field notes were recorded by the interviewer after each interview. Please see p. 8, lines 209-211.

-Who did the transcription of the audio recordings? Was it the RA or one of the authors? If it was not the RA, then how did you integrate the field notes into the transcripts?

Interviews were sent to a Professional Transcription Service used by the first author. 

We have provided additional information as to how field notes and transcripts were integrated. Please see p.9, lines 226-229.

---

## [Editor Report · Decision Letter 1]

12 Mar 2023

Accessing mental health services for a child living with anxiety: Parents’ lived experience and recommendations

PONE-D-23-01298R1

Dear Dr. Woodgate,

We’re pleased to inform you that your manuscript has been judged scientifically suitable for publication and will be formally accepted for publication once it meets all outstanding technical requirements.

Kind regards,

Nelsensius Klau Fauk, S.Fil., M., MHID, MSc, PhD

Academic Editor

PLOS ONE
---

## [Editor Report · Acceptance letter]

15 Mar 2023

PONE-D-23-01298R1 

Accessing mental health services for a child living with anxiety: Parents’ lived experience and recommendations 

Dear Dr. Woodgate:

I'm pleased to inform you that your manuscript has been deemed suitable for publication in PLOS ONE. Congratulations! Your manuscript is now with our production department. 

Kind regards, 

on behalf of

Dr. Nelsensius Klau Fauk 

Academic Editor

PLOS ONE